# Piezoelectric Nanomaterials Activated by Ultrasound in Disease Treatment

**DOI:** 10.3390/pharmaceutics15051338

**Published:** 2023-04-26

**Authors:** Shiyuan Yang, Yuan Wang, Xiaolong Liang

**Affiliations:** Department of Ultrasound, Peking University Third Hospital, Beijing 100191, China

**Keywords:** ultrasound, piezoelectric nanomaterials, therapy

## Abstract

Electric stimulation has been used in changing the morphology, status, membrane permeability, and life cycle of cells to treat certain diseases such as trauma, degenerative disease, tumor, and infection. To minimize the side effects of invasive electric stimulation, recent studies attempt to apply ultrasound to control the piezoelectric effect of nano piezoelectric material. This method not only generates an electric field but also utilizes the benefits of ultrasound such as non-invasive and mechanical effects. In this review, important elements in the system, piezoelectricity nanomaterial and ultrasound, are first analyzed. Then, we summarize recent studies categorized into five kinds, nervous system diseases treatment, musculoskeletal tissues treatment, cancer treatment, anti-bacteria therapy, and others, to prove two main mechanics under activated piezoelectricity: one is biological change on a cellular level, the other is a piezo-chemical reaction. However, there are still technical problems to be solved and regulation processes to be completed before widespread use. The core problems include how to accurately measure piezoelectricity properties, how to concisely control electricity release through complex energy transfer processes, and a deeper understanding of related bioeffects. If these problems are conquered in the future, piezoelectric nanomaterials activated by ultrasound will provide a new pathway and realize application in disease treatment.

## 1. Introduction

To date, numerous medical technologies have been developed, yet few can accommodate the diverse range of clinical applications. As a result, there is an urgent need to explore innovative and effective disease treatment strategies [1]. The employment of nanomaterials and nanotechnologies has garnered significant attention in recent decades, as various nanomedicines have demonstrated substantial success in enhancing therapeutic efficacy while reducing side effects [2,3,4]. Among these, piezoelectric materials have emerged as promising candidates for medical applications due to their unique piezoelectric properties, which enable stress-induced electric activation and can be triggered by external mechanical sources.

First discovered by P. Curie and J. Curie in 1880, piezoelectricity refers to the ability of certain materials to convert vibrational energy into an electric field in response to stress [5], a phenomenon later defined as the piezoelectric effect. Conversely, when an electric field is applied to a piezoelectric material in the polarization direction, the material deforms, releasing mechanical energy, and the inverse piezoelectric effect occurs [6]. Subsequent research over the following two centuries has elucidated the mechanisms of piezo-catalysis, employing screening charge effects and energy band theory [7].

Materials possessing the piezoelectric effect are capable of converting mechanical stress into electrical energy, thereby stimulating cells or interfering with their surrounding electric field. Consequently, piezoelectric materials can influence electron transport in vivo, acting on various physiological tissues such as stem cells, muscles, neurons, and embryogenesis, and interfering with their physiological activities. Presently, invasive electrodes are employed to apply −10 to −90 mV of electricity to alter the membranes of living cells. Following the modification of the endogenous electric field, ions such as Ca^2+^, Na^+^, and K^+^ move directionally, controlling cell excitation or inhibition (particularly in neurons) directly or by adjusting protein/ion channel conformation and expression, ultimately regulating signal pathways, and manipulating gene expression or neurotrophin levels [8].

Another mechanism involves the initiation of electrochemical reactions by electricity, impacting cells and biological processes [9]. The piezoelectric effect can induce effective interfacial charge transfer, thereby promoting the exceptional redox catalytic activities of piezoelectric materials. Numerous electrons and holes can be released from piezoelectric materials to catalyze redox reactions of water and other substrates when exposed to mechanical energy [10,11]. This process subsequently generates reactive oxygen species (ROS) that directly oxidize nearby biomolecules, damaging cancer cells or sterilizing tissues without penetration limitations [12,13,14].

Ultrasound is a well-established noninvasive and nonionizing diagnostic technology, operating within a frequency range of 2.5 to 15 MHz under diagnostic conditions. The mechanical and thermal effects of ultrasound at around 1 MHz are frequently utilized in various treatments. Additionally, ultrasound has the ability to increase the permeability of the gastric mucosa and blood–brain barrier [15,16]. Certain advantages of ultrasound, such as noninvasiveness and mechanical effects, enable its use in the controlled delivery and release of drugs.

In contrast to traditional therapies that rely on electrical stimuli and require invasive percutaneous electrodes or transcutaneous devices, wireless treatment methods employing piezoelectric nanoparticles represent a new paradigm, as they are activated by external ultrasound (US). The combination of piezoelectric nanoparticles and US allows for wireless induction of local electrical stimulation within the body [17], circumventing issues of infection and biocompatibility associated with invasive electrodes. Consequently, electrical stimuli can be directly transmitted to target tissues with high spatial precision via an in vitro wireless mechanical trigger, such as US equipment, harnessing the piezoelectric effects of nanoparticles [18]. As a result, piezoelectric materials hold promise for improving electrical stimulation treatments activated by the mechanical effects of ultrasound.

In light of the rapid advancements in piezoelectric nanomaterials activated by ultrasound, there is a pressing need for a systematic review of this domain. This review aims to address this requirement by providing a comprehensive synthesis of the most recent studies on piezoelectric nanomaterials activated by ultrasound. Initially, the discussion will encompass piezoelectric materials, nanomaterials, and their associated bioeffects, followed by an examination of ultrasound and piezoelectric phenomena. Subsequently, an extensive summary will be presented, detailing the properties and applications of piezoelectric nanomaterials activated by ultrasound in various biomedical fields, including the treatment of nervous system disorders, musculoskeletal tissue therapies, cancer treatments, antibacterial therapies, and more. Finally, the review will delve into the primary challenges faced by piezoelectric nanomaterials activated by ultrasound and offer insights into their future prospects. The objective is to facilitate the translation of this technology into clinical applications.

## 2. Piezoelectric Materials, Piezoelectric Nanomaterials, and Their Bioeffects

### 2.1. Piezoelectric Materials

Piezoelectric materials exhibit two distinct responses to external stimuli: the direct piezoelectric effect, which involves the generation of electrical energy under mechanical pressure, and the reverse piezoelectric effect, characterized by mechanical deformation caused by electrical stimulation. A fundamental equation can be employed to describe the direct piezoelectric effect as follows:(1)D=dT+εE

In this equation, *D* represents the electric displacement, *d* and *T* denote the piezoelectric coefficient and the applied stress, respectively, while *ε* and *E* correspond to the material’s dielectric constant and the electric field, respectively [19]. The piezoelectric coefficients of common piezoelectric materials are provided in Table 1 [20].

Piezoelectric materials can be fundamentally classified into three categories: inorganic, organic, and composite materials. Inorganic piezoelectric materials primarily consist of piezoelectric crystals and ceramics. Piezoelectric crystals were the earliest studied materials, dating back to the 1940s. The underlying mechanism involves external force altering the net dipole from zero to a nonzero value, which leads to a distinct electrically neutral arrangement, culminating in a positive center and a negative center and ultimately resulting in the material’s electrical polarization [21,22]. Piezoelectric crystals can be single crystals, such as SiO_2_, Rochelle salt, potassium dihydrogen phosphate (KDP), ammonium dihydrogen phosphate (ADP), or polycrystalline, including BaTiO_3_ (BTO), lead zirconate titanate (PZT), and PbTiO_3_ (non-ferroelectric piezoelectric) [23]. Piezoelectric ceramics are compounds formed after heating, with each tiny part polarizing in the same direction under the same external force, causing the material to polarize as a whole. However, the toxicity of piezoelectric ceramics renders them unsuitable for direct use within the body. Organic piezoelectric materials also referred to as piezoelectric polymers, can be divided into synthetic polymers and natural polymers [23]. Synthetic piezoelectric polymers, such as polyvinylidene fluoride (PVDF) films, exhibit flexibility, low density, low impedance, and a high piezoelectric voltage constant (g). Furthermore, their acoustic impedance is closely aligned with air, water, and biological tissue, making them particularly suitable for the fabrication of liquid, biological, and gas transducers. Additionally, certain natural materials possess favorable piezoelectric properties, including peptides, collagen, and amino acids. As a result, these materials can be employed to transfer electricity through tissues at various stages of life, not only during embryogenesis but also in adult organisms, to regulate adaptation, development, and healing processes [18]. Composite piezoelectric materials consist of two or more types of materials combined. A typical structure involves piezoelectric materials in sheet, rod, or powder form embedded within an organic polymer-based material. Such composites possess the advantages of both piezoelectric ceramics and polymer materials and have been extensively utilized in medical research. One example is the composite of piezoelectric ceramics and polymers (PVDF or others).

In general, piezoelectric ceramics exhibit higher piezoelectric coefficients, indicating their energy conversion efficiency. Conversely, piezoelectric polymers possess lower dielectric constants, representing their charge storage capability. Piezoelectric composites amalgamate the advantages of various materials to achieve specific objectives [19,20,21,22,23,24].

### 2.2. Piezoelectric Nanomaterials

Piezoelectric nanomaterials, which can be standalone materials or composites, exhibit unique physical properties such as electric field strength and dielectric constant, biological properties such as targeting and biocompatibility, chemical properties of composition and stability, etc. These properties can be tailored and enhanced through various approaches, including covalent and non-covalent modifications, as well as through the incorporation of other functional nanomaterials [25].

When piezoelectric materials are converted into nanoparticles, their piezoelectric properties may either increase or decrease. For instance, incorporating Zn_0.25_Co_0.75_Fe_2_O_4_, ZCFO, and BTO particles into PVDF-TrFE polymers enhances physical properties, such as increased electric coefficients, reduced response lags, and improved stiffness during bone tissue repair [26].

Crystalline structure and nanoparticle size are vital characteristics of nanomaterials that need monitoring. Techniques such as X-ray diffraction (XRD), scanning electron microscopy (SEM), transmission electron microscopy (TEM), piezoelectric response force microscopy (PFM), and tunneling atomic force microscopy (TUNA) are utilized to observe piezoelectric properties, while density functional theory (DFT) can be employed for predicting and understanding the material behavior [27].

Regarding biocompatibility, although certain piezoelectric materials such as lead zirconate titanate (PZT) contain toxic Pb, making their use as implantable materials contentious [28], improved biocompatibility can be achieved by coating the PZT surface with titanium [29]. Inorganic nanoparticles must acquire biocompatibility through coating before in vivo use [30], whereas organic polymers inherently possess biocompatibility. The evaluation of biosafety regarding nanoparticle shape, size, and interaction with the bio-environment is always necessary.

### 2.3. Piezoelectric Nanomaterials Bioeffects

As mentioned in the introduction, piezoelectric nanomaterials produce electricity to stimulate cells or tissues, influencing cell growth, development, proliferation, and other behaviors by affecting ions and electro-sensitive proteins. Additionally, piezoelectricity can generate reactive oxygen species (ROS) promoted by piezoelectrically polarized charges. The generated ROS induces apoptosis through oxidative stress and mediated DNA damage, making piezoelectric nanomaterials widely utilized in antitumor and pathogen clearance [31].

#### 2.3.1. Direct Electrical Stimulation

Electrical stimulation, a noninvasive and nonpharmacological physical stimulus, has extensive biomedical effects. It has been employed for muscle rehabilitation, treatment of movement/consciousness disorders, drug delivery, and wound healing. At the molecular level, electrical stimulation (ES) facilitates biomolecule transport through biofilms via electrophoresis and electro-penetration. At the subcellular level, electrical stimulation interacts with the cytoskeleton, membrane proteins, ion channels, and various intracellular organelles, altering cellular activities and functions such as contraction, migration, orientation, and proliferation [32].

#### 2.3.2. Free Radicals Based on the Piezoelectric Effect

In addition to the direct application of electrical stimulation, external mechanical forces can separate the charges of piezoelectric nanomaterials and generate ROS that impact biological systems [33]. Under mechanical vibrations, piezoelectric materials establish dynamic built-in electric fields, continuously separating electron and hole pairs for piezoelectric catalytic redox reactions, generating reactive oxygen species (ROS) such as toxic hydroxyl groups (•OH) and superoxide radicals (•O^2−^). Due to the nanosize effect, piezoelectric nanomaterials generally exhibit superior piezoelectric effects compared to macroscopic bulk structures [18].

## 3. Ultrasound in Piezoelectric Effects

The term “ultrasonic wave” refers to a mechanical wave whose frequencies exceed 20 kHz. US is widely known for its diagnostic applications, but therapeutic US has only recently become available as a safe, wireless means of inducing beneficial bioeffects within the body. There are two main physical effects caused by US waves interacting with biological tissues: thermal and mechanical effects. These effects are closely related to the tissue properties, which may vary from tissue to tissue because of their different attenuation coefficient or percentage of air content, and also related to the parameters of the ultrasound, involving US intensity, duration time, duty cycle, etc. The probability of incepting mechanically induced bioeffects is indicated by the mechanical index (MI). Additionally, piezoelectric nanoparticles can be mechanically activated by US waves.

The interaction between ultrasound and piezoelectric materials is described in a few basic models. A model put forward by Marino et al. is shown in Equation (2) [34]. The results were confirmed by an electro-elastic model of barium titanate nanoparticles (BTNP) stimulated by US.
(2)$$\varphi_{R}=-\frac{R({se}_{rr}+2e_{r\theta })}{{s\varepsilon }_{rr}}(\frac{P_{US}}{s\gamma +2\alpha })$$

In the equation, a radius is denoted by R, the piezoelectric coefficient is marked by e_rr_ and e_rθ_, and the dielectric constant is marked by ε_rr_. R, γ, and s are all known expressions, depending on the elastic properties of the BTNP. According to the equation, the voltage generated at the surface (φ) is positively correlated with the spherical particle radius R and the US wave pressure (PUS). In the presence of a US stimulus of 0.8 W/cm^2^, BTNP with a diameter of 300 nm generated about 0.19 mV of voltage.

## 4. Application of Piezoelectric Nanomaterials Activated by Ultrasound in Medicine

### 4.1. Nerve System

When neurons are at rest, their membranes are polarized, with higher levels of potassium ions inside the membrane than outside, and the opposite distribution of sodium ions. During this stage, potassium ions are highly permeable to the cell membrane, and their efflux forms a resting potential. Upon effective stimulation, sodium ion channels on the cell membrane open, and a large influx of sodium ions generate action potentials that rapidly propagate throughout the cell, creating bioelectricity that excites neurons. When an action potential reaches a synapse, presynaptic axons release neurotransmitters to postsynaptic neurons, binding to receptors in downstream dendrites and translating chemical signals into bioelectrical signals [35].

Thus, by directly modulating nerve membrane depolarization and threshold potentials, electric fields can regulate neural activity in both the peripheral and central nervous systems, resulting in cellular excitation or inhibition. For instance, deep brain electrical stimulation offers potential treatments for various nervous system disorders (e.g., pain, dystonia, essential tremor, and Parkinson’s disease) [36]. However, current electrotherapy is invasive, necessitating the insertion or implantation of electrodes, and adjusting the electrical stimulation intensity to achieve desired therapeutic outcomes is challenging and inconvenient, leading to various clinical complications [37,38]. Consequently, noninvasive treatments utilizing piezoelectric materials combined with ultrasound, which converts mechanical pressure to electricity, could provide a breakthrough.

Electrical signals can be remotely and safely delivered to excitable cells, such as neurons, using ultrasound-activated piezoelectric nanomaterials (nanoparticles or thin films) (shown in Table 2). Due to their small size, piezoelectric nanoparticles can easily enter into cells to play their function. Among related studies, Ciofani et al. first combined ultrasound with boron nitride (BN) nanotubes to polarize the nanotubes and deliver electrical stimulation to cells. Compared to the control group, this treatment caused neural-like cells (PC-12) to exhibit a 30% increase in neurite sprouting and enhanced differentiation [34,39]. This work offers new ideas and approaches for further exploring the roles of other neural cells and the functional mechanisms of stem/progenitor cells. Rojas et al. prepared barium titanate nanoparticles (BTNPs) and co-cultured them with SHSY5Y cells. After 24 h, BTNPs demonstrated a strong tendency to adsorb onto the cell membrane and extensively distributed across the membranes of SHSY5Y cells, significantly facilitating the electrophysiological response of in vitro neuronal networks when subjected to ultrasound stimulation [40].

In investigating the activation mechanism, Chen et al., through measuring the number of spines, calcium transients, and recovery time, confirmed that piezoelectric BaTiO_3_ nanoparticles can wirelessly activate neurons using ultrasound [41]. Building upon this research, they developed BTNPs decorated with a carbon shell and demonstrated its ultrasound-activated functionality in PC-12 neuron-like cells. Ultrasound-stimulated BTNPs not only increased Ca^2+^ influx but also upregulated synaptophysin and tyrosine hydroxylase levels. Additionally, they found that ultrasound-stimulated BTNPs could improve neurobehavioral disorders in zebrafish [42].

The combination with magnetic nanoparticles has resulted in further related studies. In a study conducted by Liu et al., magnetic Fe_3_O_4_ nanoparticles and piezoelectric BaTiO_3_ nanoparticles were coated on the surface of a Spirulina platensis-based micromotor. Under an effective low-intensity magnetic field, the nanoparticles could be controllably navigated to neural stem-like PC12 cells. Integrated piezoelectric BaTiO_3_ micromotors were capable of in situ electrical stimulation via the application of an external ultrasound field, further inducing differentiation in neural stem-like PC12 cells (Figure 1) [43].

More recent studies have reported additional applications, such as blood–brain barrier opening and ventricular rate reduction by stimulating the inferior right ganglionated plexus. T. Kim et al. [16] synthesized a multifunctional system consisting of NO donor N, N′-dibutyl-N, N′-dinitro-1, 4-phenylene diamine (BNN6), and piezoelectric nanoparticles barium titanate (BTNP) adhered with polydopamine (pDA). The resulting BTNP–pDA–BNN6 nanoparticle was able to locally release NO upon ultrasound irradiation, subsequently facilitating its accumulation into the brain through the blood–brain barrier (BBB) opening and response to ultrasound within the brain. This polarization caused neuron stimulation by opening voltage-gated ion channels. Moreover, BTNP–pDA–BNN6 also released dopamine from dopaminergic neuron-like cells. J. Han et al. prepared barium titanate nanoparticles (BTNPs) and injected them into adult male beagle dogs. Under ultrasound, the BaTiO_3_ piezoelectric nanoparticles reduced the ventricular rate in both sinus rhythm and atrial fibrillation models after applying ultrasound stimulation to the right inferior ganglionated plexus, compared to the control group. This suggests that ultrasound-mediated BTNP neuromodulation may be a potential treatment for atrial fibrillation rate control [44].

In addition to nanoparticles, piezoelectric films are also widely utilized. Genchi et al. designed a strategy combining poly (vinylidene fluoride-trifluoroethylene, P(VDF-TrFE)/BTNP films, and ultrasound to stimulate neurons based on the direct piezoelectric effect of P(VDF-TrFE)/BTNP films. This piezoelectric film significantly enhanced SH-SY5Y cell differentiation and induced Ca^2+^ transients [45]. Hoop et al. discovered that PVDF scaffolds with remarkable piezoelectric properties exhibited the ability to induce neuronal differentiation of PC-12 cells when exposed to ultrasound, which was similar to the induction level of nerve growth factor [46].

Modifying piezoelectric materials can enhance their effects on promoting nerve differentiation. Zhang et al. designed a novel FeOOH/PVDF membrane, based on the piezoelectric properties of PVDF, to achieve the differentiation of rat bone marrow mesenchymal stem cells (rBMSCs) under dual electrical and iron ion stimulation without any neural inducing factors involved. Both the local electrical signal generated by ultrasound-driven PVDF and the iron release controlled by piezoelectric potential acceleration are the primary reasons for the differentiation and development of rBMSCs into neurons [47].

Interestingly, ultrasound-mediated piezoeffect could also maintain the stemness of NSCs. Lu et al. synthesized poly (L-lactic) nanofibrous membrane (PLLA) with electrical signals generated through the piezoelectric effect to facilitate the stemness maintenance of NSCs. This finding contradicts previous studies that demonstrated piezoelectric signals promoting neural differentiation in other types of stem cells, such as mesenchymal stem cells. The authors further confirmed through bioinformatics and experiments that this may be related to the disruption of the Wnt pathway. These results suggest that PLLA-mediated electrical signaling may inhibit the differentiation of NSCs by negatively regulating the Wnt signaling pathway, ultimately maintaining the stemness of NSCs [48].

In addition to directly stimulating nerves through electrical signals via the piezoelectric effect, the piezoelectric effect, inducing redox reactions, may also play a crucial role in nerve systems. Qian et al. devised a boron nitride nanosheet-functionalized polycaprolactone channel scaffold (BNNS@PCL). It was demonstrated that BNNS@PCL increased electrical signal transmission and promoted the secretion of neurotrophic factors by cells to repair neurons under ultrasound stimulation in vitro. BNNS@PCL also regulated the concentrations of oxygen (O_2_) and ROS in the tissue via microcurrent. By decreasing the over-expression of ROS, BNNS@PCL restored homeostasis of the cellular internal environment and balanced O_2_ and energy metabolism, ultimately enhancing Schwann cell viability. The results showed that this approach could effectively induce the regeneration of neuronal microvasculature. Furthermore, in vivo experimental results revealed that BNNS@PCL was capable of reversing severe sciatic nerve defects and muscle atrophy that occurred after denervation. However, this research did not explore the mechanism behind ROS reduction between materials but only generalized the phenomenon to conclusions [50].

In the nervous system, ultrasound-mediated piezoelectric nanomaterials, in addition to interacting with neurons, can also target abnormal metabolites through reactive oxidative species. Jang et al. [49] discovered that piezoelectric bismuth oxide (BiOCl) nanosheets they prepared possessed the capability to dissociate Alzheimer’s disease β-amyloid (Aβ) aggregates through ultrasound-triggered redox reactions. BiOCl nanosheets exhibit piezoelectric catalytic properties, which facilitate the delivery of charge carriers into the reactants upon ultrasound stimulation, leading to redox reactions and the generation of reactive oxidative species. Consequently, the ultrasonically activated BiOCl effectively disassembled self-assembled Aβ fibrils into minute spherical fragments (Figure 2).

Therefore, electric fields generated by activated piezoelectric nanomaterials can directly induce changes in intracellular signaling pathways, neuronal activity, and ion transmission, subsequently influencing cell proliferation, differentiation, and migration, and ultimately reprogramming various cells. Piezoelectric nanomaterials can effortlessly convert ultrasonic energy into electrical energy due to their energy transformation properties, enabling them to not only modulate neural activities through electrical signals but also play crucial biological roles by catalyzing redox reactions. Therefore, they represent a highly promising tool for remote, noninvasive, and precise neural modulation.

### 4.2. Musculoskeletal Tissues

Bone, cartilage, and muscle, all of which have a highly collagenous structure, possess a piezoelectric helical structure [51,52]. Numerous bioactive scaffold piezoelectric materials have been developed for tissue engineering and regeneration. The surface charge density of piezoelectric materials can vary in response to mechanical stress, promoting bone formation and altering the physiological electrical microenvironment. By enhancing electron transfer, influencing biological signaling pathways, and increasing the production of effector molecules, the healing of musculoskeletal tissue is improved [53,54]. Ultrasound, as a remotely triggered mechanical force, plays a vital role in this process. The following sections summarize the applications in three fields: muscle, bone, and cartilage (Table 3).

#### 4.2.1. Muscle

In recent years, skeletal muscle tissue engineering (SMTE) has made great progress in reconstructing skeletal muscle not only in vitro but also in vivo. Different approaches have been reported on this subject. Biophysical stimulation during cell culture in vitro represents a talented avenue to gain more efficient skeletal muscle tissue development [70].

With the advancement of tissue engineering research, electrical stimulation (ES) has been demonstrated to play an important role in promoting the differentiation of skeletal muscle cells. By providing electrical signals, the expression of transcription factors regulating the differentiation of skeletal muscle can be stimulated [55].

Ricotti et al. [55] and Danti et al. [56], building upon previous tissue engineering technology, utilized boron nitride nanotubes (BNNT) and ultrasound-induced piezoelectric effects to promote the differentiation of C2C12 myoblasts into viable myotubes and the internalization of BNNT for further skeletal muscle differentiation.

Moreover, Paci et al. [57] developed a printable alginate/pluronic bioink containing piezoelectric BT nanomaterials for 3D bioprinting of muscle cell-loaded hydrogels. Single-layer structures embedding C2C12 cells were bioprinted.

When the cells were subjected to three consecutive days of ultrasound stimulation during differentiation, greater differentiation of skeletal muscle was observed in the cells surrounded by the doped printed constructs compared to control groups.

#### 4.2.2. Bone

Bone cells include osteocytes, osteoclasts, osteoblasts, and osteogenic cells. When injured, bones must activate specific functions of different cells for repair. This activation can be stimulated by electricity, which is mediated through the piezoelectric effect of ultrasound or other mechanical stimulations.

In this section, we focus on bone repair induced by ultrasound-mediated piezoelectric effects, particularly through direct piezoelectric effects. It is important to note that low-intensity pulsed ultrasound (LIPUS) waves can also stimulate bone tissue; however, one method of mechanical stimulation of bone tissue is via ultrasound transmission. Despite the incomplete understanding of the mechanism by which mechanical stimulation affects bone tissue, previous studies by Azuma et al. have demonstrated accelerated fracture healing rates when LIPUS was applied at any stage of the bone healing process [71]. Consequently, to account for this effect, all studies included a control group that did not receive LIPUS treatment.

Piezoelectric nanosystems, such as nanoceramics (e.g., BNTTs), can be mechanically activated by ultrasound (US) after being absorbed by osteoblasts in vitro. These activated nanomaterials have demonstrated the ability to stimulate bone extracellular matrix (ECM) formation through the upregulation of TGF-β, a factor sensitive to electrical signals [58]. Moreover, Ma et al. synthesized polynylon-11 nanoparticles (NPs), which, when irradiated with US, effectively stimulated osteogenic differentiation of dental pulp stem cells (DPSCs) in a non-invasive manner. Consequently, piezoelectric nylon-11 NPs combined with ultrasound held significant potential for tissue engineering applications, particularly in noninvasively stimulating stem cells [59].

Das et al. developed a piezoelectric nanofiber of poly (L-lactic acid) (PLLA) biological scaffold, which, when stimulated by ultrasound in vitro, facilitated remote electrical stimulation without the need for batteries [60]. The biodegradable PLLA materials possess a degradation time compatible with the osteogenesis process, allowing for controlled timing and quantity of electrical signals generated by ultrasound. In vitro experiments promoting osteogenic differentiation of stem cells revealed a positive correlation between piezoelectric signals and osteogenic differentiation. Furthermore, a mouse skull defect model demonstrated that the combination of piezoelectric material and ultrasound yielded the most effective bone defect repair (Figure 3). Yang et al. [61] observed similar effects by incorporating barium titanate into PLLA. In this study, a selective laser-sintered PLLA scaffold was produced using graphene and barium titanate (BT), which, when irritated by ultrasound, generated a current of approximately 10 nA and a high output voltage of 1.4 V. Cell experiments confirmed that the electrical signals significantly enhanced cell proliferation and differentiation.

Polyvinylidene difluoride (PVDF) has been extensively investigated as a scaffold material. P(VDF-TrFE)/boron nitride nanotube (BNNT) was fabricated utilizing the casting annealing method. P(VDF-TrFE)/BNNT film and ultrasound exposure significantly promoted the differentiation of SaOS-2 cells. Transcriptional-level studies of osteogenic differentiation marker genes revealed that composite membrane expression was upregulated after ultrasound stimulation, providing substantial evidence for a biological response induced by the direct piezoelectric effect of the film upon ultrasound exposure [62]. In a separate study, polydopamine-functionalized BaTiO_3_ nanoparticles were integrated into PVDF scaffolds fabricated through a selective laser sintering process. Upon ultrasound stimulation, the β-phase portion of the PDF/1p-BT scaffold increased by 11%, resulting in greater charge generation and a 356% surge in output voltage. Enhanced electrical stimulation more effectively promoted cell adhesion, proliferation, and differentiation [63]. Chen et al. refined the material and introduced a novel preparation of a PVDF piezoelectric film containing carbon nanotubes (CNTs) and graphene oxide (GO) additives. The proper accumulation of the β-phase augmented PVDF nanoparticles, and a positive correlation was established between the g_33_ (straight stretch) value of piezoelectric properties and the β-phase content in PVDF. Under ultrasound stimulation, the proliferation and mineralization of D1 cells cultured on the piezoelectric membrane of the GO group (0.8%) and the CNT group (0.2%) were superior to those without nanoparticles. The obtained results laid the foundation for the potential relevance of enhanced PVDF piezoelectricity in the biomedical field [64].

In addition to the aforementioned materials, BaTiO_3_-coated titanium scaffolds, which can generate piezoelectric signals through low-intensity pulsed ultrasound (LIPUS), may also play a crucial role in bone defect repair. Cai et al. [65] reported that a BT piezoelectric ceramic coating was produced on the surface of TC4 titanium alloy, forming a BT/TC4 material. Upon LIPUS stimulation, MC3T3-E1 cells exhibited osteosynthesis (cell adhesion, proliferation, and osteogenic differentiation, accompanied by calcium influx). Concurrently, J. Chen et al. [66] successfully prepared a uniform piezoelectric BaTiO_3_ coating on 3D-printed porous titanium alloy scaffolds in situ, obtaining similar results. Piezotherapy under ultrasound irradiation generated an induced current of 10–17.5 μA, which participated in the cell cycle, promoted cell viability, and reduced damaged apoptosis. The underlying mechanism may involve activating mitochondria, enriching the extracellular matrix, and upregulating the expression of osteogenesis-related genes (particularly BMP-2) in BMSCs on BaTiO_3_-coated scaffolds. Liu et al. [67] coated BaTiO_3_ on porous Ti6Al4V and applied LIPUS to evaluate the piezoelectric effect on large bone defect treatment in vitro and in vivo. The piezoelectric effect was found to stimulate osteogenic differentiation of BMSCs in vitro, as well as osteogenesis and implant growth in vivo.

Delving further into the osteogenesis mechanism, Wu et al. [68] reported that low-intensity pulsed ultrasound (LIPUS) stimulation of piezoelectric effects in polarized BaTiO_3_/Ti (BT/Ti) frameworks promoted M2 polarization of macrophages and immune-regulated osteogenesis of MC-3T3 osteoblasts. RNA sequencing and mechanistic investigations revealed that the piezoelectric BT/Ti (polarized) scaffold downregulated the inflammatory MAPK/JNK signaling cascade while stimulating oxidative phosphorylation (OXPHOS) and ATP synthesis in macrophages. These results provide a plausible mechanism for piezoelectric stimulation in promoting immunomodulatory bone repair.

In summary, poly (L-lactic acid) (PLLA) and polyvinylidene difluoride (PVDF) were employed as scaffold materials, with the addition of piezoelectric materials. In vitro and in vivo experiments confirmed that electrical currents could be generated through the piezoelectric effect under ultrasound stimulation. While the precise effects of charge production on different cells remain incompletely understood, a wealth of evidence suggests their functional and physiological roles in promoting bone repair.

#### 4.2.3. Cartilage

Chu et al. designed an innovative ultrasound-generating device capable of uniformly stimulating mesenchymal stem cells (MSCs) coated on quartz plates. Their research demonstrated that piezoelectric stimulation within an ultrasound intensity range of 1 to 20 mW/cm^2^ could encourage MSC aggregation, thereby fostering chondrogenesis of MSCs without the necessity of differentiation media [69].

### 4.3. Anticancer

While advances in cancer treatment have generally led to increased survival rates and lower cancer fatalities, trends can vary by region and cancer type. For instance, in China, the age-standardized mortality rate for cancers has decreased overall, but the incidence and mortality rates for certain cancers, such as colon–rectum, prostate, female breast, cervix, and thyroid, have increased significantly [72]. Current treatment methods, such as surgery, radiotherapy, chemotherapy, immunotherapy, and adjuvant therapy, have made clinical progress in anti-tumor efficacy. However, patients often experience severe toxic and side effects during treatment [73]. The application of nanomedicine, driven by advances in nanotechnology and materials, may address these issues. Functional and smart nanomaterials have been developed to enhance anticancer efficacy, reduce costs, and facilitate treatment. Ultrasound-activated piezoelectric nanomedicines have gained attention for their excellent tissue penetration, biocompatibility, and targeting properties. In some instances, piezoelectric nanodrugs can effectively and radically treat tumors at a lower cost than current clinical methods while inhibiting tumor recurrence with minimal adverse reactions, making them a promising cancer treatment strategy.

The mechanism of piezoelectric nanodrugs activated by ultrasound for cancer therapy can be summarized into 2 kinds: (1) based on the direct piezoelectric effect, generating electrical signals to act on cells, thereby promoting cancer cell apoptosis; (2) generating ROS through piezo-catalyzing for intracellular oxidation–reduction (redox) reactions to directly eliminate cancer cells. These are reviewed separately below (Table 4).

#### 4.3.1. Transmission of Electrical Signals

According to the literature, excessive cancer cell proliferation can be inhibited by moderate electrical stimulation [91,92]. Moreover, electrical stimulation may also enhance the efficacy of chemotherapy in brain cancer [93], which has been approved by FDA for the therapy of glioblastoma multiforme, and more clinical trials are underway [94,95].

The suppression of cell proliferation due to abnormal mitosis caused by chronic electrical stimulation can impact cancer cell proliferation without chemotherapeutic drugs and can also suppress tumor multidrug resistance [92]. However, the challenge is that normal cells are also subjected to electrical stimulation, interfering with their normal proliferation. Thus, ultrasound-mediated piezoelectric effects can achieve controlled electrical stimulation, offering a broad range of potential clinical applications.

Recent studies have demonstrated that various cancer cell types can be successfully stimulated remotely through the synergistic use of piezoelectric nanoparticles and ultrasound. Cellular studies utilizing barium titanate nanoparticles (BTNPs) revealed that chronic electrical stimulation arrested the tumor cell cycle at the G0/G1 phase by disrupting Ca^2+^ homeostasis and rectifying K^+^ channels through the upregulation of genes encoding Kir3.2. Simultaneously, chronic piezoelectric stimulation affected the organization of cytoskeletal elements involved in cell mitosis. Moreover, chronic piezostimulation increased patients’ sensitivity to chemotherapy, as demonstrated in both breast cancer cells SK-BR-3 [74] and glioblastoma multiforme cells U87 [75]. Recently, Racca et al. discovered that the combination of ZnO nanocrystals and high-energy ultrasound shock waves exhibited remarkable cytotoxicity against cervical adenocarcinoma cells, although the role of piezoelectric effects in cell death was unclear in the study [96].

In addition to direct electrical modulation, electrical modulation has been reported to be combined with other effects in tumor therapy. Pucci et al. proposed drug-loaded organic piezoelectric nanoparticles for ultrasound-responsive anti-tumor activity. The nanoplatform encapsulated nutlin-3a within ApoE-functionalized nanoparticles, which can be remotely activated by ultrasound irradiation to trigger drug release and deliver in situ anticancer electrical signals. Chemotherapy combined with chronic piezoelectric stimulation simultaneously activated cell apoptosis and anti-proliferation pathways, resulting in cell death, tumor metastasis inhibition, and reduced cell invasion in chemo-resistant GBM cells. These findings offer a less invasive and more focused approach to treat GBM and reduce drug resistance using innovative piezoelectric nanomaterials [76].

Zhan et al. designed internal electrical stimulation with varying intensities for three different BTNPs under ultrasound excitation and identified a potential mechanism by which electrical stimulation inhibited the proliferation and migration of MDA-MB-231 cells. Their research suggested that internal wireless electrical stimulation (ES) could cause both mechanical destruction and ROS bursts in cells, perturbing F-actin in the cytoskeleton and further activating related cell signaling pathways to reduce cell adhesion ability, with promising in vitro and in vivo effects (Figure 4) [77].

Alterations in tumor blood vessels induced by the direct piezoelectric effect play a significant role in cancer treatment. Li et al. [78] developed a therapeutic strategy employing ultrasound stimulation to normalize tumor vasculature using piezoelectric nanomaterials. They fabricated P-BTO nanoparticles, which exhibited a high mechanical–electrical conversion efficiency and demonstrated that these nanoparticles can generate wire-free electrical stimulation in response to LIPUS at a nanoscale interface. The results indicated that electrical stimulation could inhibit endothelial cell migration and differentiation by downregulating the eNOS/NO pathway, likely due to the disruption of the intracellular calcium gradient caused by the electrical stimulation. In vivo experiments demonstrated that electrical stimulation normalized tumor vessels by reprogramming the vascular structure, resulting in decreased vascular leakage, enhanced blood perfusion, and restored local oxygenation. Ultimately, the antitumor efficacy was substantially improved, 1.8 times greater than that of single-agent chemotherapy, through the combination of P-BTO and ultrasound. In another study, Sen et al. [79] reported a polymeric piezoelectric nanoparticle comprising apolipoprotein E (ApoE)-functionalized polymers loaded with the chemotherapeutic drug nutlin-3a (ApoE-Nut-PNPs), which could serve a dual purpose of chemotherapy and mild piezoelectric stimulation under ultrasound. ApoE-Nut-PNPs were shown to respond remotely to ultrasound irradiation, inducing antiangiogenic effects by inhibiting angiogenic growth factors, leading to the disrupted formation of tumor tubular vessels and ultimately resulting in reduced cell migration and invasion.

In addition to reducing chemotherapy resistance and modulating tumor angiogenesis, the piezoelectric effect can be integrated with immunotherapy. Kong et al. [80] utilized noninvasive local signals generated by piezoelectric β-polyvinylidene fluoride (β-PVDF) membranes under ultrasound irradiation to modulate macrophage polarization, thereby enhancing macrophage inflammatory responses and suppressing tumor cell growth. This approach relied on the micro-vibration of the β-PVDF membrane and the local charge release upon exposure to ultrasound irradiation, promoting Ca^2+^ influx through voltage-gated channels and stimulating macrophage M1 polarization and inflammatory cytokine secretion via the Ca^2+^-CAMK2A-NF-κB signaling pathway. This study revealed that electrical conduction triggered by piezoelectric materials can be controlled to drive inflammatory responses by adjusting non-invasive ultrasound doses, offering a novel method to regulate immune cell fate and support cancer immunotherapy.

#### 4.3.2. Catalyzing ROS Production

Cancer cells exhibit higher oxidative stress due to elevated ROS levels compared to normal cells. Consequently, cancer cells have been proven to be more sensitive than normal cells to drugs that further increase oxidative stress [97]. Numerous studies have demonstrated that cellular redox homeostasis can be easily disrupted by modulating antioxidant levels and increasing ROS concentrations, leading to severe cell damage and even cell death. This mechanism has been exploited to disrupt electron transport chains in tumor cells and interfere with cellular metabolic processes [98]. Eradicating cancer cells through ROS generation is a promising approach for cancer treatment, as ROS can completely destroy various cellular components critical for cell functions, such as enzymes, DNA molecules, and lipid membranes. This may potentially bypass tumor cell drug resistance and overcome the most significant obstacle of chemotherapy [99].

Owing to this well-established mechanism, piezoelectric nanomaterials have garnered extensive interest for efficiently catalyzing ROS generation upon ultrasound exposure, particularly as ultrasound-induced piezoelectric catalysis offers advantages such as deep tissue penetration, excellent spatiotemporal controllability, and in situ ROS generation [100].

For example, Biswas et al. [101] synthesized TiO_2_-BaTiO_3_ nanorods that demonstrated a remarkable capacity to kill cancer cells by generating ROS through piezoelectric catalysis under ultrasound irradiation. Li et al. [38] constructed multilayered black phosphorus (BP) nanosheets to establish the necessity for piezoelectric materials to possess an appropriate band structure to produce significant ROS under ultrasound stimulation. This finding suggested that BP nanomaterial could serve as effective sensitizers for tumor sonodynamic therapy by enabling the ultrasound-responsive eradication of cancer cells. In this study, ultrasound waves generated mechanical strain on the BP nanosheets, leading to piezopolarization, which resulted in a conduction band (CB) edge that was more negative than BP’s O_2_/•O^2−^ redox couple potential and a valence band (VB) edge that was more positive than the H_2_O/•OH redox couple potential. This process significantly accelerated ROS production. Zhu et al. prepared tetragonal BaTiO_3_ (T-BTO), another type of piezoelectric nanomaterial, which generated ROS, including toxic •OH and •O^2−^, via piezoelectric catalysis upon exposure to ultrasound. This catalytic activity downregulated the proliferation markers of Ki-67 and induced cytotoxic effects, effectively eliminating tumor cells in situ [81].

Wang et al. targeted the hypoxic tumor by developing an ultra-small barium titanate nanoparticle coated with lipid molecule DSPE-PEG2000 (P-BTO) capable of killing triple-negative breast cancer cells through a piezoelectric effect. As depicted in Figure 5, the ultra-fine P-BTO yielded unbalanced surface charges when irradiated by ultrasound, which initiated redox reactions and simultaneously generated O_2_ and ROS. O_2_ production can significantly alleviate hypoxia in the tumor microenvironment (TME) and downregulated the protein expression of HIF-1α, which was closely associated with cancer growth, angiogenesis, and metastasis. In addition, the generated ROS showed an excellent ability to rapidly cause severe damage to tumor cells with significantly reduced tumor volume compared with the control groups [13].

Glutathione (GSH) plays an essential role in modulating the intracellular redox balance. Elevated GSH levels within cancer cells can pose challenges to various cancer treatment methods. Combining piezoelectric effects with glutathione depletion has emerged as a strategy for cancer treatment. Heat-treated natural sphalerite nanosheets (NSH700 NSs) have been reported to induce ROS burst and GSH depletion within tumor cells due to their piezo-photocatalytic effect. Upon ultrasound and light irradiation, NSH700 NSs can form a polarized electric field, making charge recombination difficult, which is advantageous for efficiently generating ROS, including •O^2−^ and •OH, as well as depleting redox GSH. The superior ability of ROS production and GSH depletion ensured the excellent performance of NSH700 NSs for killing cancer cells both in vitro and in vivo, with significantly inhibited tumor growth after one treatment [82]. Furthermore, Dong et al. [83] synthesized piezoelectric nanosensitizers, ultrathin 2D Bi_2_MoO_6_-poly (ethylene glycol) nanoribbons (BMO NRs). Notably, BMO NRs can deplete endogenous GSH to disrupt redox homeostasis and be activated by GSH to form GBMO, which possessed the ability to delay charge recombination and generate more ROS under ultrasound irradiation.

Wang et al. [84] demonstrated, for the first time, that the charge generated by piezoelectricity can enhance the peroxidase-like activity of MoS_2_. The designed acidic tumor microenvironment (TME)-responsive H_2_O_2_ self-supplied cascade nanocatalysis (BTO/MoS_2_@CA) exhibited potent tumor suppression based on ferroptosis induced by redox homeostasis imbalance, enhancing the binding of MoS_2_ and H_2_O_2_, resulting in continuous •OH production to directly kill cancer cells. When combined with GSH depletion, BTO/MoS_2_@CA exhibited a substantial in vivo anti-tumor effect under ultrasound.

The anti-tumor effect can also be achieved by delivering drugs that promote oxidation or by modulating tumor metabolism in synergy with the piezoelectric effect. Recently, Hoang et al. prepared Au-decorated ZnO nanorods (Au@P-ZnO NRs) with potent reactive oxygen species (ROS) generation capabilities. They co-loaded Au@P-ZnO NRs and a pro-oxidant drug, piperlongumine, to enhance oxidation, resulting in improved therapeutic outcomes for breast cancer [102]. They subsequently designed a novel piezoelectric two-dimensional (2D) WS_2_ nanosheet for cancer treatment via piezodynamic cancer therapy triggered by ultrasound. The 2D WS_2_ nanosheets were first functionalized with triphenylphosphine (TPP), a mitochondrial-targeting molecule, to significantly potentiate their anti-cancer efficacy via a mitochondria-targeted piezodynamic therapy strategy. Furthermore, the WS_2_ nanosheets can also encapsulate FX11, a potent cellular energy metabolism inhibitor, to synergistically improve piezodynamic cancer therapy [85].

In addition to the aforementioned studies, piezoelectric effects can also be combined with other therapies, such as chemotherapy, radiotherapy, sonodynamic therapy, and photodynamic therapy, representing a novel strategy for efficient therapeutics.

The lack of lymphatic vessels within tumor tissues results in fluid retention between the tissues, creating a back pressure gradient between the tumor tissues and blood vessels. This hinders the penetration of drugs into deep tumor tissues, leading to tumor recurrence and metastasis. Consequently, Y. He et al. [86] devised a novel strategy based on piezo-catalytic nanomedicine to decrease tumor interstitial fluid pressure (TIFP) through water cleavage within the tumor stroma. First, they loaded the chemotherapeutic drug doxorubicin (DOX) into the piezoelectric nanomaterial MoS_2_ to obtain MD. Subsequently, the tumor cell membrane (CM) was coated onto MD to produce MD@C. MD@C exhibited exceptional targeting ability to tumor tissues through homologous targeting, and it could also trigger piezoelectrocatalytic water cleavage under external ultrasound irradiation to reduce pressure. As a result, MD@C significantly decreased the TIFPs of U14 and PAN02 tumors by 57.14% and 45.5%, respectively, and enhanced the tumor inhibition rate of U14 and PAN02 tumors to 96.75% and 99.21%, respectively, likely due to the deep penetration of DOX. Furthermore, hydroxyl radicals generated by piezo-catalysis in conjunction with doxorubicin effectively inhibited tumor growth.

For similar reasons, as nitric oxide (NO) is a safe and effective therapeutic technique for enhancing drug penetration by regulating the dense fibrotic matrix in the tumor microenvironment (TME), Wang et al. proposed an ultrasound-responsive nanomedicine, CPT-t-R-PEG2000@BaTiO_3_ (CRB). This nanomedicine encapsulated piezoelectric barium titanate nanoparticles (BaTiO_3_) with an amphiphilic prodrug molecule, consisting of the chemotherapeutic drug camptothecin (CPT) linked by a thioketal bond (T) and a NO donor, L-arginine ^®^. Owing to the ultrasound-triggered piezoelectric catalysis technology, BaTiO_3_ can generate ROS even in the hypoxic tumor microenvironment (TME) to initiate a cascade reaction process, breaking the thioketal bond and oxidizing R to release CPT and NO, respectively, and simultaneously delivering CPT and NO to the deep part of the tumor site. Additionally, the controlled release behavior of CPT and NO both in vitro and in vivo was observed in response to ultrasound stimulation, which promoted the depletion of the tumor dense matrix by NO, thereby enhancing the delivery and efficacy of CPT. Animal experiments demonstrated that the production of NO following ultrasound stimulation significantly improved the tumor penetration ability of CPT. The underlying mechanism was that NO can freely diffuse to the deep part of the tumor and reduce the extracellular matrix, thus remarkably inhibiting chemotherapy resistance and enhancing anti-tumor efficiency (Figure 6) [87].

Liao et al. developed a piezoelectric catalyst (BiOCl@PAA) based on BiOCl to achieve the synergistic dual catalytic effect of piezoelectricity and radioactivity. H_2_O_2_ was first generated in situ by sonicating BiOCl@PAA, and then converted into the more reactive •OH under X-ray irradiation. These novel combined therapies further expanded the biomedical applications of piezoelectric nanomaterials and demonstrated their versatile applicability [88].

Zhao et al. designed Cu_2-x_O-BaTiO_3_ nanocubes (Cu_2-x_O-BTO NCs) as a potential treatment for breast cancer. Cu_2-x_O-BTO NCs exhibited enhanced sonodynamic therapy (SDT) and synergistic chemodynamic therapy effects on breast tumors. Figure 7 demonstrated that BaTiO_3_, a classic piezoelectric material, can effectively catalyze the production of singlet oxygen (^1^O_2_) and hydroxyl radicals (•OH) by forming a built-in electric field and continuously accumulating electrons and holes on the particle surface upon exposure to ultrasound. Furthermore, the Cu_2-x_O-BTO fabricated heterojunctions facilitated the separation and movement of electrons and holes, thereby accelerating reactive oxygen species (ROS) production. Additionally, the Cu(I) in Cu_2-x_O-BTO NCs possessed catalytic activity to induce Fenton-like reactions, allowing the conversion of abundant endogenous H_2_O_2_ within tumor tissues into more oxidative and lethal •OH radicals to directly damage cancer cells [89]. In this study, the coordination of SDT and chemodynamic therapy was achieved using piezoelectric heterostructures as both sonosensitizers and chemical kinetic agents. This strategy significantly improved cancer treatment efficacy and provided a valuable reference for designing novel piezo-catalysts.

Apart from chemotherapy, radiotherapy, and sonodynamic therapy, piezoelectric nanoparticles can also be combined with photodynamic therapy. Ramedani et al. developed a novel liposome nanoparticle composed of an anticancer drug (silibinin), photoluminescent materials (graphene quantum dots, GQDs), and a piezoelectric polymer (P(VDF-TrFE)) for the diagnosis and treatment of breast cancer. In vivo studies confirmed that the administration of nanoparticles and application of external stimuli (light and ultrasound) dramatically inhibited 4T1 tumor growth and reduced the volume of 4T1 breast cancer in BALB/c mice. Decorated with PD-1 antibodies, the nanoparticles selectively targeted breast cancer tumors and exhibited minimal distribution in other normal tissues, thereby enhancing the theranostics of breast cancer. These combined properties of nanoparticles provided an appealing platform for effective photodynamic therapy and cancer imaging [90].

### 4.4. Antibacterial Therapy

Researchers have been striving to develop novel antibacterial materials. However, some of these materials face challenges related to biotoxicity, efficacy, and stability, limiting their use in medicine. It is crucial to establish a novel, effective, and simplified mechanism to prevent bacterial growth [103]. Piezoelectric materials can generate positive and negative charges on their surface upon ultrasonic excitation, which interferes with the normal physiological activities of bacteria. Simultaneously, they catalyze the redox reaction between water molecules and oxygen to produce reactive oxygen species (ROS), which affects the electron transport chain of bacterial metabolism, thereby damaging bacterial cell walls and denaturing bacterial proteins [104,105,106]. When bacteria are exposed to sufficient oxidative stress, membrane destruction ensues. During membrane disruption, toxic substances such as aldehydes and oxides are produced within the bacteria. These products are believed to function as secondary reactive messengers, which can transform into destructive forms of internal proteins, DNA, and other nucleophiles. We summarize some of the studies below (Table 5).

Vatlin et al. first reported the preparation and characterization of piezoelectric polyhydroxybutyric acid, polyvinylidene difluoride, polyvinylidene difluoride trifluoride, and non-piezoelectric polycaprolactone polymer films, and the significant inhibition of bacterial growth by the mechanical stimulus of piezoelectric materials through ultrasound treatment [107]. Thus, the present findings demonstrated a specific antibacterial effect of piezoelectric polymers in various tissue engineering applications.

Gazvoda et al. [112] prepared an all-organic piezoelectric biodegradable film composed of poly (L-lactide) (PLLA) and demonstrated its antibacterial activity against Gram-positive bacterial model Staphylococcus epidermidis and Gram-negative bacterial model Escherichia coli. Comparisons between nanotextured surfaces with and without piezoelectric properties ruled out a major role for morphology and directly confirmed that piezoelectricity was the primary cause of the observed antibacterial effects. The study also demonstrated that with an antibacterial nanofiber membrane, piezoelectric stimulation could directly disrupt the bacterial membrane. This was the primary mechanism of action, and the contribution of pH changes and ROS production was negligible. More importantly, this effect was selective for bacterial membranes, and the same damage did not occur in human erythrocytes, making it suitable for further therapeutic use.

In addition to the direct antibacterial effect through the piezoelectric effect, most studies have achieved antibacterial effects through the generation of reactive oxygen species catalyzed by the piezoelectric effect, with some used in combination with sonodynamic therapy. The design of new materials with wound healing–promoting functions is beneficial because piezoelectric materials can promote the proliferation and migration of fibroblasts under ultrasound stimulation, thereby accelerating wound healing [113].

Liu et al. [108] developed a multifunctional hydrogel based on ultrasound-triggered piezo-catalytic therapy to promote healing of bacterial-infected wounds. Under ultrasonic vibration, reactive oxygen species (ROS) were rapidly generated on the surface of BaTiO_3_ (BT) nanoparticles embedded within the hydrogel due to the powerful built-in electric field, endowing the hydrogel with superior antibacterial properties. The hybrid hydrogel confined the BT nanoparticles within the wound site, locally inducing piezoelectric catalysis to eliminate bacteria under ultrasound, thereby significantly enhancing the biosafety and bioavailability of therapeutic procedures. Similarly, Lei et al. [12] introduced an appropriate amount of oxygen vacancies into BTO through sulfur doping, yielding the resulting SDBTO with improved piezo-catalytic ROS production performance. The present study demonstrated that SDBTO could successfully heal tibial bone defects and suppress the inflammatory response in rats infected with Staphylococcus aureus. Triggered by medical ultrasound, the nanomaterial generated piezoelectric signals and promoted osteogenic differentiation of hBMSCs through the TGF-β signaling pathway to repair bone defects.

Li et al. [109] proposed a rapid and effective treatment strategy for osteomyelitis using ultrasound-triggered BiFeO_3_/MXene (Ti_3_C_2_) ferroelectric polarization interface engineering. Under ultrasound, ferroelectric polarization led to the establishment of a piezoelectric field. Ultrasound cavitation effect-induced sonoluminescence also caused BiFeO_3_/Ti_3_C_2_ to generate photogenerated carriers. BiFeO_3_/Ti_3_C_2_ accelerated the separation of electrons and holes while inhibiting electron backflow under the synergistic action of the polarization electric field and Schottky junction, resulting in higher utilization of polarized and photogenerated charges, and subsequently higher ROS production under ultrasound conditions. Consequently, more ROS were produced to effectively eradicate bacteria. Likewise, Wu et al. [114] designed a piezoelectric nanodrug (Au@BTO NCs) composed of BaTiO_3_ nanocubes loaded with Au nanomaterials. Upon ultrasound irradiation, BaTiO_3_ nanoparticles underwent mechanical deformation, generating electron–hole pairs that separated and moved in opposite directions under the Schottky barrier with Au atoms. The resulting surface charge carriers further activated water and oxygen molecules to generate •OH and ^1^O_2_. Both in vitro and in vivo experiments demonstrated the nanomaterial’s potent antibacterial capabilities, as well as its promotion of fibroblast and macrophage migration and repair of infected wounds.

Combining pH-stimulated drug delivery with ultrasound-controlled sonodynamic therapy may offer a multifunctional treatment for septic wounds. Zhu et al. (Figure 8) [110] designed dynamically evolving antibacterial and repair-promoting nanocomposites (NCs) by in situ assembly of zeolitic imidazolate framework-8 (ZIF-8) on the barium titanate (BTO) surface and subsequent loading with a small amount of ciprofloxacin (CIP). The system can dynamically control the amount of ROS produced at different wound recovery stages, effectively balancing antiseptic and pro-repair effects during the treatment process and presenting a novel direction for designing effective multifunctional platforms for biomedical applications.

### 4.5. Others

Apart from that, piezoelectric nanoparticles can eliminate abnormal deposits that accumulate in neurons and can also be used for arthritis treatment. Excessive reactive oxygen species (ROS) generated by aberrant mitochondria is one of the important causes of rheumatoid arthritis (RA). The sonocatalytic two-dimensional piezoelectric nanosheets Fe/BiOCl prepared by Li, et al. can efficiently create electrons under ultrasound motivation to meet the purpose of depleting mitochondrial outer membrane H (+) and interfering with mitochondrial matrix H (+) supply. This led to mitochondrial membrane potential (MMP) depolarization, triggering mitophagy in the inflamed region to eliminate the source of ROS regeneration [111].

## 5. Perspectives and Conclusions

Piezoelectric materials, by designing into various drugs, can be thin films and can be nanoparticles, under mechanical action to produce an electric current, through direct electrical signals or oxidation-reduction reactions, in particular, ROS to achieve the corresponding biological effects. This article mainly reviews the piezoelectric effect stimulated by ultrasound, which has been applied in many fields of medicine, including the nervous system, musculoskeletal system, tumor, and antibacterial. Yet in the process of application, there are still the following problems: (1) how to measure piezoelectric properties; (2) how to determine the US dose which can generate a piezoelectric effect in vitro and vivo; (3) how to ensure the related bioeffects.

### 5.1. How to Measure Piezoelectric Properties

Piezoelectric properties are important properties of piezoelectric materials and, although more complex to measure, are important to select the most appropriate material for a certain biomedical application, likewise, to support precise analysis or arithmetic models. As shown in Table 2, Table 3, Table 4 and Table 5, most of the previous studies did not measure the piezoelectric properties of the new materials. Maybe people are not so concerned with the piezoelectric constant but are very concerned with the application, however, this affects further exploration of the application mechanism. If performed correctly, it can actually help to properly understand and quantify the physics of piezoelectric effects mediated by ultrasound.

Piezoelectric response force microscopy (PFM) has been discovered as a bright technique to reveal and quantify the piezoelectric properties of nanoparticles since 1992 [115]. Recently, with further exploration, this technique has become a standard characterization tool at micro- and nano-scales [116]. PFM is a special scanning probe microscopy mode that uses alternating current (AC) voltage to conclude the piezoelectric coefficient of a material. The standard PFM model utilizes the indirect piezoelectric effect. The use of direct effects to extrapolate electrical signals from materials has only recently made some progress [117]. Nanomaterials with complex geometries can be measured using PFM due to their high resolution and nondestructive imaging capabilities.

So far, we still cannot identify a simple and completely correct method to quantify the piezoelectric coefficients of nanosubstance. Although PFM represents an invaluable platform to study the piezoelectric properties of materials, the tool has not yet been incorporated into a standardized evaluation tool for industrial applications. Standardized methods will be necessary to ensure the consistency and reliability of PFM measurements performed in different laboratories. The establishment of a quantitative identification protocol will ensure the homogenization process of the detection methods of various laboratories in the future.

### 5.2. How to Determine the US Dose That Can Generate Piezoelectric Effect In Vitro and In Vivo

The ultrasound dose required to activate piezoelectric nanoparticles is a crucial influence factor. Some studies did not report the ultrasound dose used, while others employed ultrasonic baths, leading to unpredictable exposure conditions since researchers can only adjust the device’s electrical power. As demonstrated in Table 2, Table 3, Table 4 and Table 5, these factors make it challenging to establish a quantitative relationship between the ultrasound dose and specific effects. In some investigations, various frequencies and intensities were considered to determine the appropriate dose for the experiment. However, due to missing or limited information on ultrasound sources and acoustic transmission, the findings may not be comparable or reproducible by other researchers [118].

The optimal distance between the ultrasound probe and biological targets is often overlooked. In the “near field” (i.e., distances below D2/4λ, where D = transducer diameter and λ = wavelength), placing the sample too close to the ultrasound probe results in significant fluctuations in the spatial distribution of the ultrasound intensity. To obtain optimal results, the sample should be positioned in the “far field,” ensuring better uniformity of ultrasound in terms of local intensity distribution [119].

Real-time in situ monitoring of ultrasound-induced physical effects should be performed to exclude other phenomena that could overlap with electrical ones and to directly confirm and control efficacy and safety in vivo [18].

In in vivo, it is essential to note that sound attenuation varies among different tissues. For deep tissue locations, reflection, attenuation, diffraction, and other physical phenomena can influence and distort the ultrasound beam. Adequate acoustic coupling between the probe and the human body is crucial. If not performed correctly, energy transfer through the body can dramatically alter the activation dose in the target tissue [120].

### 5.3. How to Ensure the Related Bioeffects

In fact, the mechanisms underlying cellular responses to piezoelectric stimulation have not been fully elucidated. The involved pathways may be influenced by various factors and conditions, such as different cell types and varying degrees of stimulation required to produce piezoelectric effects. Ultrasound can induce thermal and cavitation effects, which may overlap with piezoelectric effects and collectively impact cell behavior, potentially leading to misinterpretation of results. Thus, it is crucial to conduct experiments with multiple condition controls and appropriate control groups.

As mentioned in the former parts, several opposing findings were noted on different cell types. For instance, piezoelectric motivation on one hand can foster the proliferation of fibroblasts [114], and macrophages [80], and on the other hand, can hinder cell cycle progression in diverse malignant cells, such as breast carcinoma cells [74], and glioblastoma multiforme cells [75], yet these disparate cellular replies are only ostensibly paradoxical.

In fact, similar to piezoelectric stimulation, direct electrical stimulation is also known to facilitate fibroblast proliferation [121] and inhibit cancer cell proliferation via cell cycle arrest and mitotic spindle disruption. It is worth noting that the biochemical pathways activated by piezoelectric and direct electrical stimulation are generally similar.

Further research on piezoelectric biological effects is necessary, particularly in different types of tissue cells and even bacteria, to develop more targeted nanomedicine.

In conclusion, the combination of piezoelectric nanoparticles and ultrasound stimulation has emerged in the past decade as a groundbreaking approach in various biomedical fields, including neuromodulation, regenerative medicine, tumor therapy, and antibacterial therapy. Despite this progress, numerous opportunities and challenges still need to be explored and addressed in these medical fields. In addition to the aspects discussed above, targeting strategies, biocompatibility studies, and quantification of US-mediated piezoelectric effects should be considered. In the near future, with rapid advancements in innovative technology and piezoelectric materials, nanoparticles with piezoelectric properties are expected to be further developed, refined, and progressively implemented in clinical settings, fostering medical innovation and improving health outcomes.

## Figures and Tables

**Figure 1 pharmaceutics-15-01338-f001:**
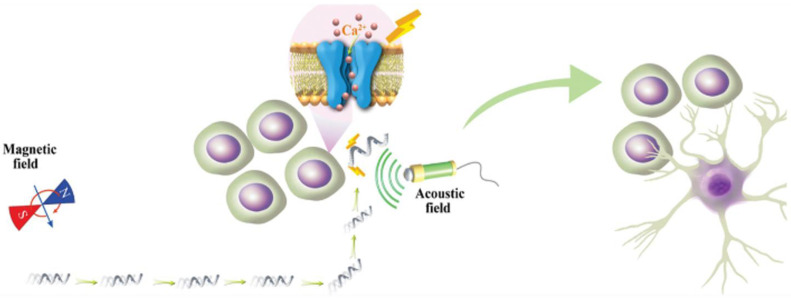
A highly manageable micromotor to differentiate the targeted neural stem-like cell [43].

**Figure 2 pharmaceutics-15-01338-f002:**
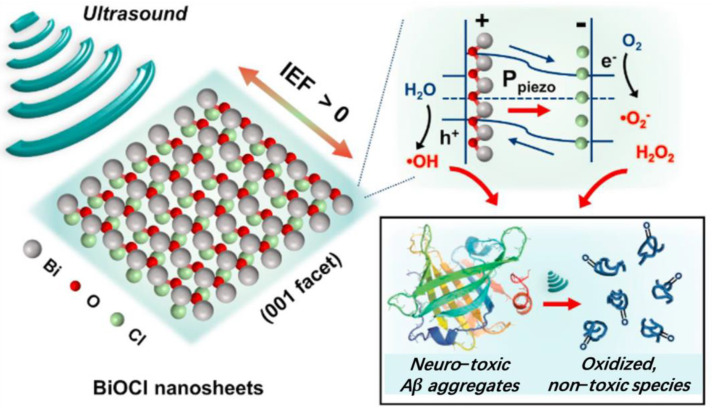
Schematic illustration of BiOCl nanosheets dissociating self−assembled Aβ fibrils of Alzheimer’s disease [49].

**Figure 3 pharmaceutics-15-01338-f003:**
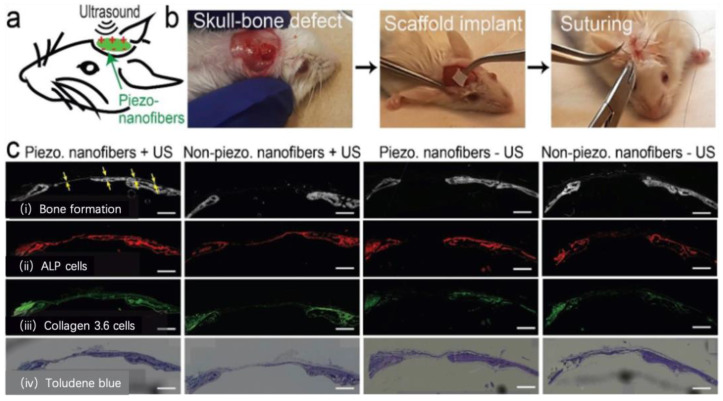
Representative histology sections of the mouse calvarial bone showing details of bone formation and cell migration into the defects for the groups of piezo-scaffold (3000 rpm) and non-piezo. scaffold (300 rpm) under the same application of US. (**a**) Is a simplified schematic of our in vivo experiment. (**b**) Shows the sequence of the different steps involved in the surgery. We created a critical sized calvarial defect in mice and implanted our nanofiber film into it to observe bone formation on the film. This is followed by treating the animals with US at the site of implantation to mimic the in-vitro system. The experiment was performed at n = 5 animals per group. (**c**) **i**. Aperture contrast images comparing the mineral formation in the defect between the four animal groups. Yellow arrows indicate the new bone formation, clearly seen in the first group while it cannot be seen in the other groups. (**c**) **ii**. Fluorescent images comparing the presence of ALP-producing cells in the defect between the four groups using vector blue ALP staining. (**c**) **iii**. Microscopic fluorescent images comparing the migration of Collagen 3.6 gene-positive cells in the defect between the four groups. (**c**) **iv**. Microscopic optical images comparing the bone formation in the defect between the four groups using Toludene blue staining. All scale bars are 1 mm [60].

**Figure 4 pharmaceutics-15-01338-f004:**
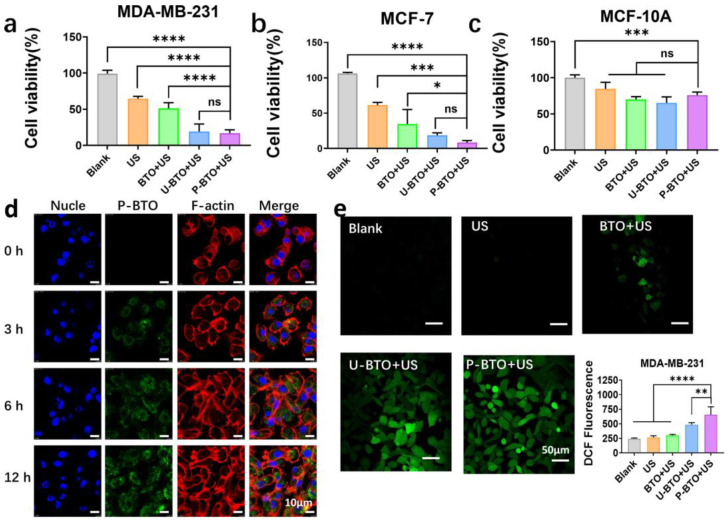
Cytotoxic and intracellular distribution of BTNPs. (**a**–**c**) Cell viability of MDA-MB-231, MCF-7, and MCF-10A after BTNP treatment for 24 h and US treatment (28 kHz, 2.25 W/cm^2^, 30 s). (**d**) Fluorescence images of MDA-MB-231 cells taking up P-BTO at different times (0, 3, 6, 12 h). Blue, DAPI-stained DNA; red, F-actin; green, P-BTO. (**e**) Fluorescence images and semiquantitative analysis indicating ROS generation in MDA-MB-231 cells after BTNP treatments with or without US. (n = 4 for each group) * indicates *p* < 0.05, ** indicates *p* < 0.01, *** indicates *p* < 0.001, **** indicates *p* < 0.0001, ns indicates *p* > 0.05 [77].

**Figure 5 pharmaceutics-15-01338-f005:**
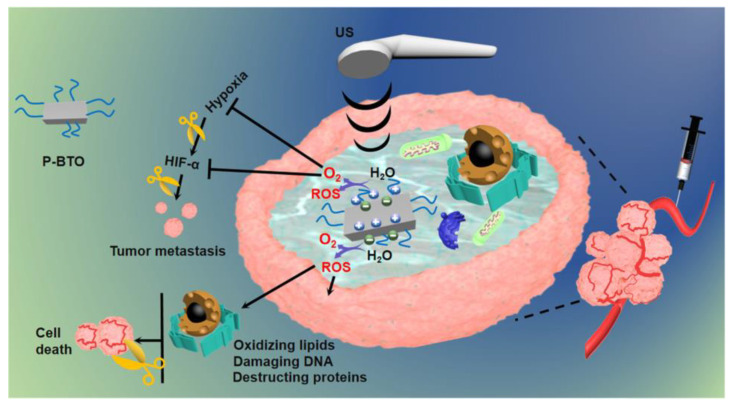
P-BTO nanoparticles combined with ultrasound can generate both O_2_ and ROS based on piezo-catalytic effect, which showed remarkable effect in tumor treatment and metastasis inhibition [13].

**Figure 6 pharmaceutics-15-01338-f006:**
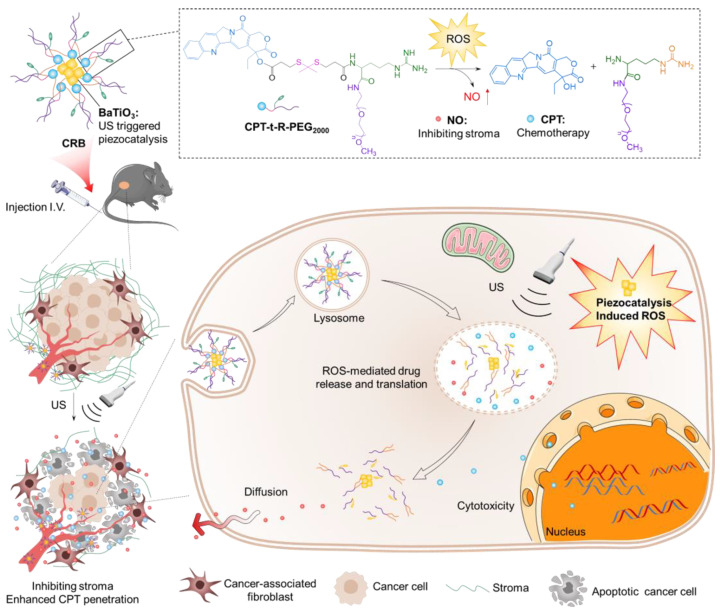
CRB nanoprodrug can release CPT and NO on-demand via US-triggered piezo-catalysis of BaTiO_3_ to enhance PAN02 tumor chemotherapy [87].

**Figure 7 pharmaceutics-15-01338-f007:**
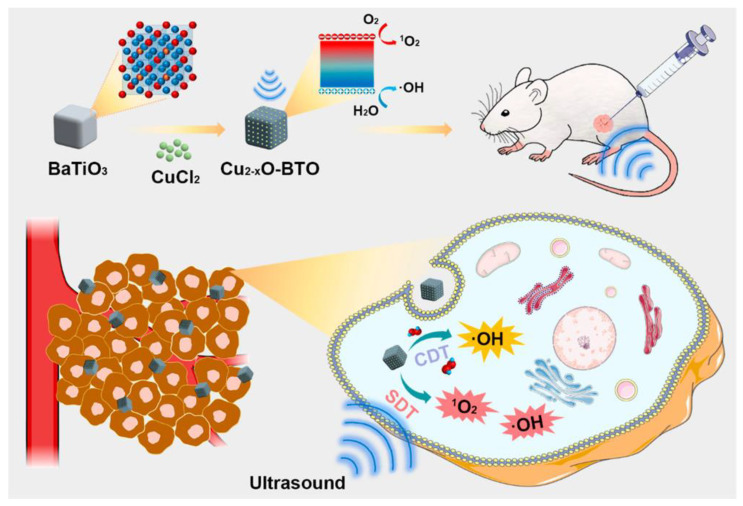
Enhanced sonodynamic and chemodynamic therapy based on Cu_2−x_O−BTO NCs [89].

**Figure 8 pharmaceutics-15-01338-f008:**
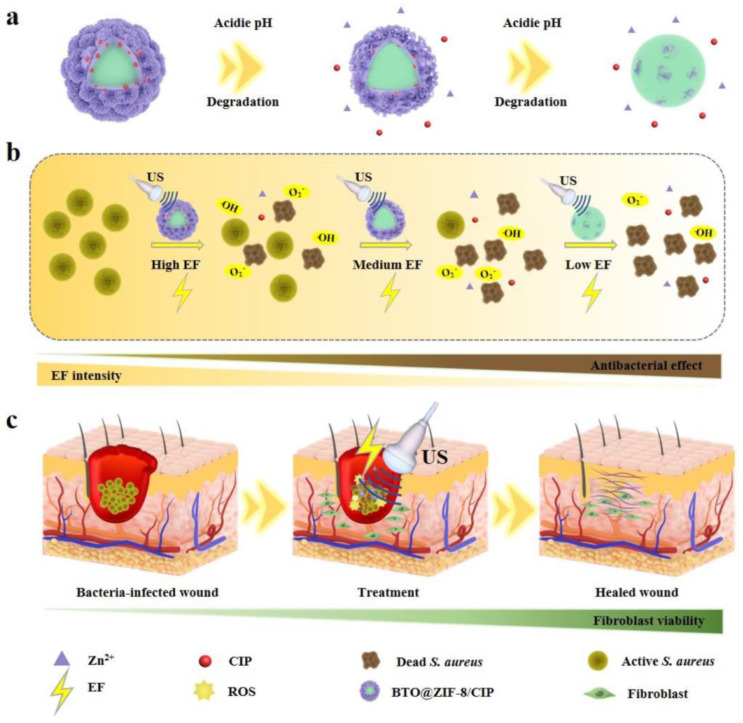
Mechanism of BTO@ZIF−8/CIP NC in antibacterial repair of tissue. (**a**) Schematic illustration of pH-stimulated decomposition of BTO@ZIF−8/CIP NC. (**b**) Schematic illustration of antibacterial mechanisms of BTO@ZIF−8/CIP NC. (**c**) Schematic illustration of wound repair promotion mechanisms of BTO@ZIF-8/CIP NCs [110].

**Table 1 pharmaceutics-15-01338-t001:** The piezoelectric coefficients of common piezoelectric materials.

Classification	Materials	Piezoelectric Charge Constant, dj (pC/N)	
		Bulk	Nanofiber
Ceramic materials	PbNb_2_O_6_	d_33_ = 57	-
	PbTiO_3_	d_33_ = 97	-
	BaTiO_3_	d_33_ = 95	-
	PBLN	d_33_ = 350	-
	Li_2_B_4_O_7_	d_33_ = 8.76	-
	PZT	d_33_ = 223	-
	ZNO	d_33_ = 12.3	-
Polymer materials	PVDF	d_31_ = 23, d_33_ = 15	d_33_ = 57.6
	PLLA	d_14_ = 6.0	d_33_ = 3±1
	PHBV	d_33_ = 0.43	d_33_ = 0.7 ± 0.5
	PAN	d_31_ = 0.6	d_33_ = 39, 1.5
	Nylon-11	d_33_ = 6.5, d_31_ = 14	-
	Collagen	d_14_ = 12, d_33_ = 2	d_15_ = 1
	Cellulose	d_25_ = 2.1	d_33_ = 31
	Chitin	d_33_ = 9.49	-
	Chitosan	d_31_ = 10, d_33_ = 4.4	-

Lead zirconate titanate, PZT; polyvinylidene fluoride, PVEF; poly (L-lactic acid), PLLA; Poly (3-hydroxybutyrate-co-3-hydroxyvalerate), PHBV; polyacrylonitrile, PAN.

**Table 2 pharmaceutics-15-01338-t002:** Application of piezoelectric nanomaterials triggered by ultrasound in nerve system.

Materials	Particle Diameter	Piezoelectric Coefficients	Ultrasound Features In Vitro	Cell Type	Cell Outcome	Ultrasound Features In Vivo	In Vivo	Vivo Outcome	Ref.
BTNP–pDA–BNN6	261.15 ± 5.14 nm	/	/	/	/	1.5 MHz, 462.4 W/cm^2^, 10% duty cycle	Male C57BL/6 mice	Opening BBB, opening voltage-gated ion channel, releasing neurotransmitter to synapse	[16]
BT NPs with gum arabic coating	479.0 ± 145.3 nm	/	1 MHz, 0.8 W/cm^2^, 5 s	SH-SY5Y	Activating voltage-gated Ca^2+^ and Na^+^ channels	/	/	/	[34]
BN nanotubes with glycol chitosan coating	length 200–600 nm, diameter 50 nm	/	40 kHz, 20 W, 5 s per time	PC-12 SH-SY5Y	Increasing neurite elongation	/	/	/	[39]
BTNPs with gum arabic coating	116.8 ± 46.5 nm	/	1 MHz, 1 W/cm^2^, 3 min	Primary rat cortical and hippocampal neurons	Increasing the neural network activity	/	/	/	[40]
BTNPs with DSPE-PEG5000 coating	30 μm	/	500 kHz, 2 kPa, 10 s	Primary rat cortex neurons	Increasing Ca^2+^ concentration and neuron network response	/	/	/	[41]
BTNPs with carbon shell	66 ± 10 nm	/	1 MHz, 0.64 W/cm^2^, 5 min, for 7 days	PC-12	Increase Ca^2+^ influx; upregulate synaptophysin and tyrosine hydroxylase	3 times (10 min each) per day for 1 week	PD zebrafish model zebrafish	Distinctly improvement after the treatment	[42]
BTNPs with gum arabic coating	0.8 μm	/	1 MHz, 1 W/cm^2^	PC-12	Activating voltage-dependent Ca^2+^ channels and adenylyl cyclase pathway	/	/	/	[43]
BTNPs	/	/	/	/	/	3 MHz at 2.75 W/cm^2^ for 5 min	Adult male beagles	Reducing the ventricular rate by stimulating the inferior right ganglionated plexus	[44]
BTNPs in (P(VDF-TrFE))	212 nm	d_31_ = 53.5 pm/V g_31_ = 0.24 mV/N	1 MHz, 1 W/cm^2^, 5s	SH-SY5Y	Enhance Ca^2+^ transients and neurite lengths	/	/	/	[45]
Polyvinylidene fluoride PVDF films	20 μm	d_33_ = −30 ± 2 pC/N	132 kHz, 80 W, 10 min, 5 times a day	PC-12	Promote the cell differentiation	/	/	/	[46]
FeOOH/PVDF	/	d_33_ = 27.2 pC/N	400 W, 8 min, twice a day	rbMSCs	Promote rbMSCs into neuron-like cells	/	/	/	[47]
PLLA	500–700 nm	/	300 W, 0.5 V	NSCs	Maintain the stemness of NSCs during proliferation	/	/	/	[48]
BiOCl	Width: 0.8–2.5 μm, thickness: 200–470 nm	/	/	/	/	49 W	5xFAD mouse model of AD	Decreased the density of amyloid plaques	[49]

**Table 3 pharmaceutics-15-01338-t003:** Application of piezoelectric nanomaterials triggered by ultrasound in musculoskeletal system.

Materials	Particle Diameter	Piezoelectric Coefficients	Ultrasound Features In Vitro	Cell Type	Cell Outcome	Ultrasound Features In Vivo	In Vivo	Vivo Outcome	Ref.
Boron nitride nanotubes	/	/	5 s, 20 W, 40 kHz	NHDF, C2C12	Promote skeletal muscle differentiation	/	/	/	[55]
Boron nitride nanotube	/	/	/	C2C12	Promote skeletal muscle differentiation	/	/	/	[56]
BTNPs	273.3 ± 10.7 nm	d_33_ = 88 pm/V	5 min, 1 MHz; 250 mW/cm^2^; 1 kHz, 20% duty cycle	C2C13	Promote skeletal muscle differentiation	/	/	/	[57]
BNNTs	40–70 nm, lengths <500 nm, thickness 10 nm		40 kHz, 5 s, 20 W	Primary human osteoblasts	Stimulating bone ECM formation by upregulation of TGF-β	/	/	/	[58]
Piezoelectric nylon-11 nanoparticles	50 nm	/		DPSCs	Promote the osteogenic differentiation of DPSCs	/	/	/	[59]
Poly (L-lactic acid) (PLLA)	/	/	40 kHz, 20 min each day and performed for 10 days.	adipose derived stem cells	Piezoelectric charge on enhanced osteogenesis	30 min per day, 5 days per week, 4 weeks in total, 40 kHz,	Mice	Repair of bone defects	[60]
PLLA/BT/graphene scaffold	400 μm		0.8 W/cm^2^, 40 KHz, 100% duty cycle, 100 Hz	hUC-MSCs	Cell proliferation and differentiation	/	/	/	[61]
P(VDF-TrFE)/BNNT	10 nm	d_31_ = 11 ± 4 pm/V, g_31_ = 0.155 ± 0.056 Vm/N, e_31_ = 7.59 ± 3.52 mC/m^2^	Twice a day for 10 s, 1 W/cm^2^, 100 Hz, 100% duty cycle	SaOS-2 osteosarcoma cells	Osteogenic differentiation	/	/	/	[62]
PVDF/p-BT	/	/	Three times a day for 10 s, 0.8 W/cm^2^, 100 Hz	MG-63 cells	Cell adhesion, proliferation, and differentiation	/	/	/	[63]
PVDF piezoelectric membrane	/	/	40 kHz, 20 min a day	D1 cells	Better proliferation and mineralization	/	/	/	[64]
BT/TC4 materials	1.6 μm	d_33_ = 0.42 pC/N	1 min, 1 MHz, with a repetition rate of 100 Hz and 30 mW/cm^2^, daily exposure 20 min	MC3T3-E1 cells	Greatest osteogenesis	/	/	/	[65]
BaTiO_3_-coating titanium scaffold	Thickness 50 μm	/	1.5 MHz, 30 mW/cm^2^, 1 kHz, 20% duty cycle, 20 min per day	BMSCs	Better viability and adhesion, increasing the expression of osteo-genesis-related genes BMP-2	/	/	/	[66]
BaTiO_3_-coating porous Ti_6_A_l4_V		d_33_ = 0.7 pC/N	1.5 MHz, 200 μs, 1 kHz, 30 mW/cm^2^, 10 min daily	BMSCs	Promote the osteogenic differentiation	1.5 MHz, 200 μs, 1 kHz, 30 mW/cm^2^, 10 min daily	Sheep	Bone formation and growth into implants in vivo	[67]
BaTiO_3_/Ti_6_A_l4_V (BT/Ti) scaffold	/	/	/	/	/	10 min, at 3, 5, 7 and 14 days, 1.5 MHz, 0.2 ms, 30 mW/cm^2^	SD rats	Decreased the ratio of M1 macrophages	[68]
AT-cut quartz coverslip	/	/	(80 W, 132 kHz) for 10 min	Mesenchymal stem cells	Drove clustering facilitated chondrogenesis	/	/	/	[69]

**Table 4 pharmaceutics-15-01338-t004:** Application of piezoelectric nanomaterials triggered by ultrasound in tumor treatment.

Materials	Particle Diameter	Piezoelectric Coefficients	Ultrasound Features In Vitro	Cell Type	Cell Outcome	Ultrasound Features In Vivo	In Vivo	Vivo Outcome	Ref.
DSPE-PEG2000 coated BaTiO_3_ nanoparticle	6.83 ± 1.75 nm	/	1 MHz, 1 W/cm^2^, 50% duty cycle, 5 min	4T1	ROS and O_2_ generation	1 MHz, 1 W/cm^2^, 50% duty cycle, 10 min	4T1 tumor-bearing mice	Downregulate HIF-1α, and ROS can kill tumor cells	[13]
Few-layer black phosphorus (BP) nanosheet	Thickness 5.3 ± 3.7 nm average lateral 162.4 ± 99.4 nm	/	0.14 W/cm^2^, 15 min	4T1	ROS generation	Once a day for the first 4 days, 1 MHz, 1.5 W/cm^2^, 10 min	4T1 tumor-bearing mice	suppressed tumor growth and metastasis without causing off-target toxicity	[38]
P(VDF-TrFE)/BaTiO_3_ nanoparticle composite films	≈212 nm	/	1 W/cm^2^ (100 Hz burst rate)	SH-SY5Y	Elicit Ca^2+^ transients	/	/	/	[45]
Barium titanate nanoparticles (BTNPs)	150 nm	/	0.2–1 W/cm^2^, 0.5 Hz, 10% duty cycle	SK-BR-3	Inhibit proliferation	/	/	/	[74]
BTNPS with anti-TfR antibody (AbBTNPs)	252 ± 11 nm	/	1 W/cm^2^, 1 MHz, 200 ms, every 2 s, 1 h per day, for 4 days.	U87	Reduce proliferation, increased sensitivity to the chemotherapy treatment	/	/	/	[75]
Nutlin-3a-loaded ApoE- functionalized nanoparticles P(VDF-TrFE)	115 ± 20 nm	/	Total 2 s, 200 ms each	98G, U251, and U87 M	reduce cell migration, actin polymerization, and invasion ability, fostering apoptotic and necrotic events	/	/	/	[76]
Barium titanate nanoparticles (BTNPs)	100 ± 47 nm	/	28 kHz, 2.25 W/cm^2^, 30 s	MDA-MB-231	Inhibited cell growth and migration up	28 kHz, 2.25 W/cm^2^, 1 min	MDA-MB-231 tumor model	Suppressing tumor growth	[77]
P-BTO nanoparticles	120.70 ± 41.48 nm	/	1 MHz, 1.0 W/cm^2^, 1 min per day	HUVECs	Inhibits endothelial cell migration and angiogenesis	1 MHz, 1.0 W/cm^2^, 10 min, every other day, 3 times in total	A375 tumor model	tumor vascular normalization and anti-tumor efficacy of doxorubicin	[78]
ApoE-Nut- PNPs	76 ± 16 nm	/	1 MHz and 1 W/cm^2^, 200 ms each and activated every 2 s	hCMEC/D3	Inhibition of angiogenic growth factors	/	/	/	[79]
*β*-phase poly(vinylidene fluoride) (*β*-PVDF) film	2 μm	d_33_ = 16.22 pC/N	80 kHz, 12.5 μs	HP-1, HeLa, HepG2	Enhanced the M1 polarization of macrophages and exerted cytocidal effects against tumor cells		/	/	[80]
tetragonal BaTiO_3_ (T-BTO)	106.91 ± 49.72 nm	/	1.0 MHz, 1.0 W/cm^2^, 50% duty cycle	4T1	Kill cancer cells	3 h, 2, and 3 days, 10 min administration	4T1 tumor model	Eradicate tumors	[81]
NSH700 nanosheets	130 nm	/	0.8 MHz, 0.5 W/cm^2,^ 50% duty cycle, 10 min	MCF 7	ROS generation	40 kHz; 3 W/cm^2^; 50% duty cycle; 5 min	MCF7 tumor model	ROS accumulation and anti-tumor effects	[82]
Bi_2_MoO_6_ nanoribbons	79.26 nm long, 19.95 nm wide, and 6.03 nm thick	/	40 kHz; 3 W/cm^2^; 50% duty cycle, 5 min	HeLa	GSH depletion to amplify oxidative stress	40 kHz; 3 W/cm^2^; 50% duty cycle; 5 min at 12, 24, and 48 h	U14 tumor model	Tumor growth suppression	[83]
BTO/MoS_2_@CA (16:1)	/	/	/	/	/	10 min per day, 3 days	CT26 tumor model	Trigger ferroptosis to cause Tumor elimination	[84]
FX11@TPEG-WS2	Lateral 100 nm, thickness 5 nm	/	1 MHz, 0.5 W/cm^2^, 2 min	MCF-7	Produce ROS, inhibition glycolysis, apoptosis	1 MHz, 0.5 W/cm^2^, 3 min	MCF-7 tumor model	Tumor growth inhibition	[85]
MD@C	200 nm	/	/	/	/	1.5 W/cm^2^, 3 min	U14 and PAN02 tumor model	increased the perfusion of blood-derived drugs and inhibit the growth of tumor	[86]
CPT-t-R-PEG2000@BaTiO_3_ (CRB)	256.6 ± 38.2 nm	/	1.0 MHz, 1.5 W/cm^2^, 50% duty, 2 min	Panc02	Release ROS, CPT, and NO	1.0 MHz, 1.5 W/cm^2^, 50% duty, 5 min	PAN02 tumor models	enhanced penetration of CPT for tumor growth inhibition	[87]
BiOCl@PAA	150 nm	/	1.0 W/cm^2^,1.0 MHz, 50% duty cycle, 1 min	4T1	Increase free radicals and H_2_O_2_	5 min, 1.0 MHz, 2.5 W/cm^2^, 50% duty cycle	4T1 tumor model	Tumor eradicated	[88]
Cu_2−x_O−BTO NCs	162.3 ± 3.5 nm	/	1.0 MHz, 1.0 W/cm^2^, 50% duty cycle, 3 min	4T1	Kill tumor cells	1.0 MHz, 1.0 W/cm^2^, 5 min, 50% duty cycle	4T1 tumor- model	SDT and CDT realized effective tumor suppression	[89]
P(VDF-TrFE), graphene quantum dots (GQDs), and Silibinin (a hydrophobic drug)	230 ± 20 nm	/	/	/	/	0.1 MPa, 1.5 MHz, 1.007 kHz, 300 mW/cm^2^	4T1 tumor model	Suppressing tumor growth	[90]

**Table 5 pharmaceutics-15-01338-t005:** Application of piezoelectric nanomaterials triggered by ultrasound in antibacterial therapy and others.

Materials	Particle Diameter	Piezoelectric Coefficients	Ultrasound Feature	Bacteria Type	In Vitro/Vivo	Outcome	Ref.
ZnO@GDY NR	40 ± 6 nm	/	1 W/cm^2^, 1 MHz and 100% duty cycle	*Multidrug-resistant pathogens*. *Staphylococcus aureus*. *Pseudomonas aeruginosa*	Rat model	Decomposition of hydrogen peroxide (H_2_O_2_) and production of reactive oxygen species, almost 100% antibacterial efficacy	[105]
PLLA nanotextured films	r = 177 ± 28 nm and l = 27 ± 2 μm	/	30 min, 80 kHz, 30% power	*S. epidermidis* and *E. coli*	In vitro	Confirmed piezoelectricity as the main reason for the observed antimicrobial affect	[107]
barium titanate (BaTiO_3_, BT) nanoparticles embedded in the hydrogel	117 ± 42 nm	/	10 min (1.5 W/cm^2^, 1 MHz)	*E. coli* and *S. aureus*	Rat model	Eradicate bacteria and accelerate full-thickness skin wound healing	[108]
BiFeO_3_/Ti_3_C_2_	500 nm	/	1.0 MHz, 1.5 W/cm^2^, 50% duty cycle	*Staphylococcus*	Osteomyelitis in vitro	Enhancing the yield of reactive oxygen species under US and ultrasonic heating for killing bacteria	[109]
BTO@ZIF-8/CIP NCs	168 nm	/	1.5 W/cm^2^, 1 MHz, 50% duty cycle, 1 min	*S. aureus*	Infected mice	pH-stimulated drug delivery and ultrasound-controlled sonodynamic offering a multifunctional therapy	[110]
Two-dimensional piezoelectric nanosheets (NSs) Fe/BiOCl	Around 55 nm, 3 nm thickness	/	1.0 MHz, 50% duty cycle, day 1 h, 2, 3, and 4, 3 min, 1.5 W/cm^2^	/	RA model mice	Alleviated RA by inducing mitophagy	[111]

## Data Availability

No data listed in the manuscript were obtained from experimental studies.

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
