# Peer review of "Piezoelectric Nanomaterials Activated by Ultrasound in Disease Treatment"

_pharmaceutics, 2023, doi:10.3390/pharmaceutics15051338_

Round 1

Reviewer 1 Report

This review article on the use of piezoelectric materials in various therapies is a useful overview, but it can be difficult to read due to so many acronyms, many of which are unnecessary, and it is disjoint in many places. To improve readability, I would suggest not using acronyms unless used at least 4 times. Additionally, several acronyms like BT and PLLA were re-introduced on different occasions. It would be best to define them once and then be consistent. 

Other points - in 2.2 it states "Piezoelectric nanomaterials are piezoelectric composites with decorations to gain physic functions like electric intensity and electric constancy, biological functions like targeting and biocompatibility, chemical functions, and other functions, through covalent and noncovalent approaches in the field of nanomaterial wrapping."  I disagree with this statement. Piezoelectric nanomaterials do not have to be composites but the converse has to be true. Also it should be physical functions and I do not understand what electric intensity is.

Line 142 - "Crystalline structure and nanoparticle size are important characteristics of nanomaterial that should be monitored. X-ray diffraction (XRD), density functional theory (DFT), tunneling atomic force microscopy (TUNA), transmission electron microscopy (TEM), scanning electron microscopy (SEM), and piezoelectric response force microscopy (PFM) are used to monitor piezoelectric properties [26]." DFT can be used to model behavior but in cannot be used to monitor properties.

Line 190 - it seems that model was put forth rather than a hypothesis. The former seeks to simulate behavior while the latter is a testable idea.

Line 462 - The statement that cancer fatalities are increasing each year is made without a reference. In general, advances in cancer treatment have led to increased survival rates and lower cancer fatalities.

Line 745 - it says "We also proved that ..." Should that be "they" instead of "we" or something completely different?

Reviewer 2 Report

The authors have collected important factors of piezoelectricity nanomaterial and ultrasound. Afterward, the authors have summarized recent studies in nervous system diseases treatment, musculoskeletal tissues treatment, cancer treatment, anti-bacteria therapy, and others under activated piezoelectricity. The authors have also provided core problems, including how to accurately measure piezoelectricity properties, how to concisely control electricity release through complex energy transfer process, and deeper understanding of related bioeffects. Overall, this review can inspire more design ideas of piezoelectricity nanomaterials for disease treatment. Therefore, I would like to recommend this work to publish in Pharmaceutics. Below are some comments for the authors.

1. In this manuscript, too many paragraphs have been composed by one sentence. This review would be more impressive if the authors could revise these paragraphs composed by one sentence.

2. For the introduction “Employing nanomaterials and nanotechnologies, various nanomedicines ....”, more references could be cited to broaden the introduction.

https://doi.org/10.2147/IJN.S328767

Reviewer 3 Report

It is a comprehensive review of piezoelectric nanomaterials, related research has been included. However, the resolution of the images is not satisfactory. The English needs to be improved significantly. These issues diminished the interest of the readers, including me.  

Round 2

Reviewer 1 Report

The authors have addressed the concerns raised.

Reviewer 3 Report

The authors addressed my concerns, image resolution has been improved, as well as the English. This manuscript is ready for publication. One more minor issue, in the Abstract, the font of two words is different, it can be corrected before publication and does not affect the merit of this work.